# Metabolomics Analysis Provides Novel Insights into the Difference in Meat Quality between Different Pig Breeds

**DOI:** 10.3390/foods12183476

**Published:** 2023-09-19

**Authors:** Hongwei Liu, Jun He, Zehong Yuan, Kunhong Xie, Zongze He, Xiang Zhou, Man Wang, Jian He

**Affiliations:** 1School of Life Science and Engineering, Southwest University of Science and Technology, Mianyang 621010, China; liuhongwei0936@163.com (H.L.); zhouxiang21wb@163.com (X.Z.); wangman5782@126.com (M.W.); 2Institute of Animal Nutrition, Sichuan Agricultural University, Chengdu 611130, China; hejun8067@163.com (J.H.); yuan5213524@163.com (Z.Y.); xiekunhong20130234@163.com (K.X.); xxzongze@gmail.com (Z.H.)

**Keywords:** Chuanzang black pig, DLY, carcass traits, meat quality, myofiber type transformation, LC-MS/MS

## Abstract

The Chuanzang black (CB) pig is a new crossbred between Chinese local breeds and modern breeds. Here, we investigated the growth performance, plasma indexes, carcass traits, and meat quality characteristics of conventional DLY (Duroc × Landrace × Yorkshire) crossbreed and CB pigs. The LC-MS/MS-based metabolomics of pork from DLY and CB pigs, as well as the relationship between the changes in the metabolic spectrum and meat quality, were analyzed. In this study, CB pigs presented lower final body weight, average daily gain, carcass weight, and eye muscle area than DLY pigs (*p* ˂ 0.05). Conversely, the ratio of feed to gain, marbling score, and meat color score of longissimus dorsi (LD) were higher in CB than DLY pigs (*p* ˂ 0.05). Moreover, psoas major (PM) showed a higher meat color score and a lower cooking loss in CB than DLY pigs (*p* ˂ 0.05). Interestingly, CB pigs showed lower myofiber diameter and area but higher myofiber density than DLY pigs (*p* ˂ 0.05). Furthermore, the mRNA expression levels of *MyHC I*, *PPARδ*, *MEF2C*, *NFATC1,* and *AMPKα1* were higher in CB than DLY pigs (*p* ˂ 0.05). Importantly, a total of 753 metabolites were detected in the two tissues (e.g., *psoas major* and *longissimus dorsi*) of CB and DLY pigs, of which the difference in metabolite profiles in *psoas major* between crossbreeds was greater than that in *longissimus dorsi*. Specifically, palmitic acid, stearic acid, L-aspartic acid, corticosterone, and tetrahydrocorticosterone were the most relevant metabolites of *psoas major* meat quality, and tetrahydrocorticosterone, L-Palmitoylcarnitine, arachidic acid, erucic acid, and 13Z,16Z-docosadienoic acid in *longissimus dorsi* meat were positively correlated with meat quality. The most significantly enriched KEGG pathways in *psoas major* and *longissimus dorsi* pork were galactose metabolism and purine metabolism, respectively. These results not only indicated improved meat quality in CB pigs as compared to DLY pigs but may also assist in rational target selection for nutritional intervention or genetic breeding in the swine industry.

## 1. Introduction

Pork, as the largest-selling meat product among land animals, has been widely favored by consumers [1,2]. For years, researchers have focused on increasing the growth rate and lean meat percentage of pigs while ignoring the deterioration in meat quality [3]. Recently, with the improvement of consumers’ economic level and health awareness, safety and quality have gradually become the core of pork consumption [4,5]. As a complex index, meat quality is evaluated not only by a series of intrinsic traits, such as pH value, flesh color, drip amount, and shear force, but also by the basic unit of skeletal muscle, namely muscle fiber type composition. Based on myosin heavy chain (MyHC) polymorphisms, skeletal muscle fibers can be classified into type I (MyHC I), type IIa (MyHC IIa), type IIb (MyHC IIb), and type IIx (MyHC IIx) [6,7,8,9]. Moreover, a higher percentage of MyHC I and MyHC IIa represents greater meat quality [6,10]. As we all know, pork quality can be influenced by many factors, such as breed, nutritional status, environmental management, and slaughtering practices [11]. Among them, genetic background plays a more decisive role than feed conditions, and improving meat quality via breeding is widely used in the pig industry [12,13].

Duroc × Landrace × Yorkshire (DLY) crossbred pigs were widely bred worldwide because of their high feed conversion, fast growth rate, and excellent lean meat percentage [2,13,14]. However, DLY crossbred pigs run the risk of producing PSE (soft, pale, and exudative) or DFD (firm, dark, and dry) meat [15,16]. According to the literature, Chinese indigenous pig breeds account for about 1/3 rd of the world’s pig breeds, which are gradually favored by consumers due to their advantages of high meat quality [15]. Nevertheless, Chinese indigenous pig breeds generally exhibit undesirable traits such as poorer growth performance, a low slaughter rate, and a low lean-to-fat ratio [17]. In this case, cross-breeding can successfully improve the growth rate and lean meat percentage of Chinese indigenous pig breeds without compromising pork eating quality [18]. As a new variety bred in 2014 in China, the Chuanzang black (CB) pig was obtained by crossing traditionally Chinese indigenous pig breeds with modern breeds (Duroc × (Berkshire × (Tibet × Meishan))), which have strong disease resistance and high fertility. Although breed and genetic variations are closely related to variations in meat quality traits, little is known about whether CB pigs possess superior meat quality compared to conventional DLY crossbred pigs.

Metabolomics based on mass spectrometry (MS) is a widespread strategy that can be used to determine and quantify numerous compounds in various muscle samples [19,20,21]. With the continuous development of omics technology, liquid chromatograph-tandem mass spectrometry (LC/MS-MS) has become an important means to detect a variety of small molecule metabolites and their correlation with meat quality attributes due to its advantages of high resolution and wide dynamic range [22]. Interestingly, meat-based metabolomic studies have confirmed that metabolomics profiles could screen out differential markers that distinguish live and dead pork [23], as well as different pig breeds [24]. Until the present, no studies have been reported on the identification of the meat-based metabolomic signature between DLY and CB pigs. Consequently, the aim of this study was to describe the influences of breed on growth performance, plasma indicators, carcass traits, and meat quality. Through the systematic study of metabolic compounds and metabolic pathways, the scientific basis of how breeds alter meat quality can be established, thereby providing new insights into breed selection for meat quality improvement.

## 2. Materials and Methods

### 2.1. Experimental Design

A total of 100 healthy pigs were included in this study, with 50 pigs (castrated boars) from each of the two crossbreeds, including 50 DLY pigs (average 60.18 ± 0.24 kg) and 50 CB pigs (average 60.28 ± 0.30 kg). Male pigs were castrated within 10 days of birth. Duroc × (Berkshire × (Tibet × Meishan)) (abbreviated as CB) is obtained by crossing the sow TM (Tibet × Meishan) with the intermediate sire Berkshire and terminal sire Duroc. The detailed information on the management and lineage of these populations has been presented in previous reports [25,26]. According to the difference in breed, all pigs were assigned to the DLY group and the CB group, with 10 duplicates and 5 pigs per replicate. All pigs were raised at an experimental farm (Sichuan Tieqilishi Industrial Co., Ltd. Mianyang, China) and maintained in the same environment. They were fed the basic diet (Appendix A) required by the Nutrient Requirements Council (NRC 2012) and were provided with clean water and feed ad libitum. The trial lasted for 42 days. Feed intake was recorded each day, and weight was measured at 0 d and 42 d to calculate the average daily gain (ADG), average daily feed intake (ADFI), and feed to gain (F/G). At the end of the experiment, one pig approaching the average body weight was selected from each duplicate. The live weight of all pigs was recorded before slaughter (average weight, DLY: 128.89 ± 0.97 kg; CB: 120.26 ± 0.49 kg). The experimental procedures presented in this study followed the regulations of the Animal Care and Use Committee of the Southwest University of Science and Technology (protocol code L2023024).

### 2.2. Sample Collection

At the end of the trial, all selected pigs were weighed after fasting for 24 h and then slaughtered in accordance with standard commercial procedures. Before slaughter, approximately 10-mL blood samples were taken from the anterior vena cava to obtain serum by centrifugation (4000× *g* for 30 min at 4 °C). The serum samples were stored at −20 °C for the following determination. Subsequently, pigs were electrocuted (300 V for 3 s). The abdomen of each carcass was immediately opened and eviscerated, which was separated along the midline. An estimated 50 g of longissimus dorsi (LD) and psoas major (PM) muscle was taken from the 10th rib of the left side carcass and immediately sent to the Novogene Bioinformatics Technology Co., Ltd. (Beijing, China) for *LC-MS/MS* metabolomics analysis. The LD muscle and PM muscle from the 10th and 16th ribs were collected to detect pH value, meat color, drip loss, cooking loss, shear force, marbling score, and muscle flexibility measurement. LD and PM muscle samples from the 10th rib of the left carcass were collected for the analysis of RT-PCR and stored at −80 °C until RNA extraction. About 5 cm of LD and PM muscle samples were fixed in 4% buffered paraformaldehyde for histomorphology measurement.

### 2.3. Serum Biochemical Parameters

The contents of serum triglyceride (TG), high-density lipoprotein cholesterol (HDL-C), low-density lipoprotein cholesterol (LDL-C), total cholesterol (T-CHO), glucose (GLU), and glycosylated serum protein (GSP) were assayed using TG (Cat. No. A110-1-1), HDL-C (Cat. No. A112-1-1), LDL-C (Cat. No. A113-1-1), T-CHO (Cat. No. A111-1-1), GLU (Cat. No. F006-1-1), and GSP (Cat. No. A037-2-1) commercial kits (Nanjing Jiancheng Bioengineering Institute, Jiangsu, China), respectively. Each index is tested in duplicate using an automatic analyzer (Olympus, Shanghai, China).

### 2.4. Carcass Traits and Visceral Indexes

Carcass weight was recorded within 5 min after slaughter in order to calculate the dressing percentage. The values of backfat depth at the first, tenth, and last lumbar of the left carcass was measured to calculate the average backfat depth. The eye muscle area (EMA) was measured at the last lumbar. The abdominal fat was weighed for the abdominal adipose index measurement. The weights of the heart, liver, kidney, pancreas, and abdominal adipose tissue were measured for visceral index analysis.

### 2.5. Meat Quality

The value of pH, color coordinates, drip loss, cooking loss, water loss, marbling score, and shear force in meat was evaluated as previously outlined [27,28]. After calibrating with pH 4.6 and 7.0 buffers, the pH value of muscle was measured at 45 min and 24 h postmortem by using a pH meter (pH-STAR, SFK-Technology, Denmark). At 45 min and 24 h postmortem, meat color (lightness L*, redness a*, and yellowness b*) was measured in triplicate with a portable chroma meter (CR-300, Minolta, Japan), which was calibrated against a white tile before use. The drip loss was calculated as the difference from the initial chop weight after 24 h. Briefly, a cube muscle with a diameter of 4 cm was weighed at 45 min postmortem and suspended in a plastic bag at 4 °C for 24 h. Subsequently, the muscle was removed from the bag and reweighed. Drip loss was determined by calculating the weight change percentage. For cooking loss determination, approximately 30 g of muscle samples were weighed, transferred to a steamer, and steamed with boiling water for 30 min. Then, the muscle was taken from the steamer, dried, and reweighed after about 20 min. The cooking loss was calculated as the weight change percentage. The water loss rate was detected by the pressure method. Briefly, a round LD muscle sample with a diameter of 5 cm^2^ and a thickness of 1 cm was taken from the left carcass. After weighing, the sample was weighed again after applying 35 kg of pressure. The water loss rate was calculated as the weight change percentage. The marbling score and meat color score were measured by a mean value calculated from three observers following the National Pork Producer Council (NPPC) standards after 24 h of storage at 4 °C. Shear force was detected with a texture analyzer (TA.XT. Plus, Stable Micro Systems, Godalming, UK). Specifically, the muscle was heated in an 80 °C water bath until the muscle core temperature reached 75 °C and then cooled at 4 °C for 24 h. For each muscle sample, the cylindrical core (1.27 cm in diameter) was parallel to the fiber orientation, and the cores were sheared perpendicular to the fiber orientation. The muscles were sheared perpendicularly to the longitudinal orientation of the muscle fibers using a texture analyzer.

### 2.6. Skeletal Muscle Histomorphology

The histomorphology of myofibers in LD and PM samples was measured as previously outlined [29]. Muscle samples were fixed with 4% paraformaldehyde, then the fixed muscle samples were embedded in paraffin medium, then paraffin was sliced to 5-μm thickness and stained with hematoxylin and eosin (HE) to calculate the diameter, amount, area, and density of muscle fibers.

### 2.7. Real-Time Quantitative PCR (RT-qPCR)

Total RNA from muscle samples was extracted by an RNAiso Plus reagent (Saiwei Biotechnology Co., Ltd, Wuhan, China) and reverse-transcribed into complementary DNA (cDNA) for RT-PCR. RT-qPCR was performed by a Bio-Rad iQ6 instrument (Bio-Rad, Hercules, CA, USA) using SYBR Master Mix (Saiwei Biotechnology Co., Ltd, Wuhan, China). Relative mRNA expression was calculated by the 2 ^−ΔΔCt^ method and normalized to *β-actin* mRNA. The primer sequences are shown in Appendix A.

### 2.8. LC-MS/MS Metabolomics Analyses

According to the difference in breed, all pigs were assigned to the DLY group and the CB group, with 10 duplicates and 5 pigs per replicate. At the end of the experiment, one pig from each duplicate that was closest to the average weight of each group was selected for slaughter. According to Li et al. [30], Longissimus dorsi and psoas major muscle samples from ten pigs in each group were sampled for *LC-MS* metabolomics analyses. Approximately 50 mg of frozen muscle was placed in 500 μL of methanol: water (4:1, *v*/*v*) and homogenized by adding 6 μg of internal standard (lidocaine). The homogenates were ultrasonically extracted in an ice bath for 30 min and then centrifuged for 10 min at 12,000× *g* at 4 °C. Subsequently, the lower 400 μL was transferred to the original centrifuge tube and enriched in V-AQ mode for 5 h using a vacuum concentrator (Eppendorf 5305). The samples were dissolved in 200 μL of isopropanol and filtered through a 0.22-μm membrane to obtain the samples, which were then transferred to an LC-MS sampling bottle with an inner liner for LC-MS analysis.

Metabolomics analysis was carried out by Novogene Co., Ltd. (Beijing, China) using an UHPLC system (ThermoFisher Scientific, Waltham, MA, USA). The separation was performed on a Hypesil Gold column (2.1 mm × 100 mm, 1.9 µm) in combination with a Q ExactiveTM HF-X mass spectrometer (ThermoFisher Scientific) in positive/negative polarity mode. The water gradient flow rate of formic acid was 0.2 mL/min. The mobile phases of 0.1% formic acid and 5 mmol/L ammonium acetate were in positive ion mode and negative ion mode, respectively. The elution gradient was as follows: eluting with 2% methanol for 1.5 min; eluting with 2-100% methanol for 12.0 min; eluting with 100% methanol for 14.0 min; eluting with 2-100% methanol for 14.1 min; eluting with 2% methanol for 17 min. Centroid data were collected in the range of 50 to 1000 *m*/*z*, the scanning time is 0.1 s, and the scanning delay is 0.02 s over 13 min.

### 2.9. Data Processing and Analyses

Raw data files were processed by filtering, identifying, integrating, correcting, aligning, and normalizing with Compound Discoverer 3.1 (ThermoFisher Scientific). The molecular formula of metabolites was predicted using additive ions, molecular ion peaks, and fragment ion data matrixes, and accurate qualitative and relative quantitative results were obtained using the mzCloud, mzVault, and MassList databases.

After data processing, the principal component analysis (PCA) model and the partial least squares discriminant analysis (PLS-DA) model in SIMCA-P14.0 software (Umetrics, Umeå, Sweden) were used to assess the data. The permutation test was employed as parameters for validating the PLS-DA model, and the goodness of fit was evaluated using R2 and Q2 values. The criteria for the identification of differentially abundant metabolites were set as follows: *p*-value ˂ 0.05 and variable importance in projection (VIP) > 1. After that, analysts can use public databases such as MassBank (http://www.massbank.jp/URL (accessed on 11 May 2023)), Human Metabolome Database (http://www.hmdb.ca/URL (accessed on 11 May 2023)), and Metlin (https://metlin.scripps.edu/URL (accessed on 11 May 2023)) to identify metabolites. The functions of these metabolites were studied using the Kyoto Encyclopedia of Genes and Genomes (KEGG) database (http://www.genome.jp/kegg/URL (accessed on 11 May 2023)). Hierarchical clustering analysis and heat map analysis were carried out using the R package v3.4.

### 2.10. Statistical Analysis

The statistical analyses of all data were performed by the SPSS 23.0 software (Chicago, IL, USA) and presented as the mean and standard error of the means (SEM). The Student’s *t*-test was used for a two-group comparison. *p* ˂ 0.10 and *p* ˂ 0.05 were used to assess statistical trends and significance between the means, respectively. The correlations between meat quality traits and muscle metabolites were determined by Pearson correlation analysis in SAS.

## 3. Results

### 3.1. Growth Performance and Serum Biochemical Indexes

The growth performance of DLY and CB pigs is shown in Table 1. The CB pigs presented a lower final weight and ADG, while F/G was higher in CB than in DLY pigs (*p* ˂ 0.05). The serum biochemical indexes of DLY and CB pigs are presented in Table 2. The concentrations of serum CHO (*p* ˂ 0.05), HDL (*p* ˂ 0.05), LDL (*p* ˂ 0.05), TG (*p* = 0.09), and GLU (*p* = 0.06) were lower in CB than in DLY pigs. However, there are no differences in the content of GSP between DLY and CB pigs (*p* > 0.05).

### 3.2. Carcass Traits and Meat Quality

The carcass characteristics and visceral indexes of DLY and CB pigs are shown in Table 3. The dressing percentage, backfat thickness, and abdominal fat index were not significantly affected by breed, but carcass length (*p* ˂ 0.05), EMA (*p* ˂ 0.05), heart index (*p* ˂ 0.05), liver index (*p* ˂ 0.05), pancreas index (*p* ˂ 0.05), and carcass weight (*p* = 0.09) were lower in CB than DLY pigs. Conversely, kidney index (*p* ˂ 0.05) was lower in DLY than in CB pigs.

The skeletal muscle histomorphology of DLY and CB pigs is presented in Figure 1. Compared with the DLY group, the CB group had decreased myofiber area and diameter and increased myofiber density in LD and PM muscles (*p* ˂ 0.05). Meat quality characteristics of the DLY and CB pigs were listed in Table 4. In LD muscle, pH _45 min_, pH _24 h_, marbling score, and meat color score values were lower (*p* < 0.05) in DLY than in CB pigs. Longissimus dorsi water loss rate values from DLY pigs tended to be higher (*p* = 0.059) than those of CB pigs. In PM muscle, pH _24 h_, pH _45 min_, and meat color score were lower (*p* ˂ 0.05) in DLY than in CB pigs. *Psoas major* L*_45 min_, L*_24 h_, b*_24 h_, and cooking loss were lower in CB than in DLY pigs. However, longissimus dorsi L*_45 min_, b*_45 min_, L*_24 h_, a*_24 h_, b*_24 h_, drip loss, cooking loss, shear force, and muscle flexibility, and psoas major a*_45 min_, b*_45 min_, a*_24 h_, marbling score, and drip loss did not attain statistical significance (*p* > 0.05).

### 3.3. Relative mRNA Levels of Myofiber Type Transformation

As shown in Figure 2A,B, the mRNA levels of *MyHC I*, *PPARδ*, *MEF2C*, and *NFATC1* were higher in LD muscle from CB pigs than those in DLY pigs. In PM muscle, CB pigs had increased (*p* ˂ 0.05) mRNA levels of *MyHC I*, *AMPKα1*, *MEF2C*, and *NFATC1*, and decreased (*p* ˂ 0.05) mRNA levels of *MyHC IIb* and *MyHC IIx* compared to the DLY pigs (Figure 2C,D).

### 3.4. Metabolic Profiles in Muscles

After alignment and data pre-processing, 753 metabolites of muscle were suitable to include in the statistical analysis. In a PCA model including DLY-LD, DLY-PM, CB-LD, and CB-PM groups (*n* = 40), the largest variation in the data [19.8% of the explanatory variation (PCA1)] was associated with the site of muscle tissue, and the second variation in the data [12.51% of the explanatory variation (PCA2)] was associated with the pig breeds (Figure 3A). Further analysis of multivariate statistical methods with supervised pattern recognition analysis via the PLS-DA model was performed to identify metabolic separation among the LY-LC, DLY-PM, CB-LD, and CB-PM groups and begin the feature selection process. Overall, metabolic separation among the LY-LC, DLY-PM, CB-LD, and CB-PM groups was observed. And the site of muscle tissue factor presents clustering patterns or trends in the PLS-DA model (35.06% of the explained variation (PC1)). Furthermore, the model is distinctly separated by the factor associated with pig breeds in the second component [19.95% of the explained variation (PC2)]. Hierarchical cluster analysis (HCA) results of the dataset showed that the samples presented modest evidence of clustering by the site of muscle tissue and pig breeds, which corresponded to overlapping populations detected by PCA and PLS-DA. Collectively, the above results showed that there are likely metabolite differences in the longissimus dorsi and psoas major muscles of DLY and CB pigs.

As shown in Figure 4, 45 up-regulated metabolites and 64 down-regulated metabolites were accurately identified in the longissimus dorsi muscle of DLY and CB pigs. Among the metabolites of longissimus dorsi muscle with a greater diversity in abundance were fatty acyl groups, including eicosapentaenoic acid ethyl ester, 13-HPODE, 13Z,16Z-docosadienoic acid, erucic acid, arachidic acid, elaidic acid, and adrenic acid. In addition, the longissimus dorsi muscle from BC pigs increased (*p* < 0.05) the contents of inosine 5′-monophosphate, cortisone, corticosterone, cortisol, ergocalciferol, and tetrahydrocorticosterone, while decreasing (*p* < 0.05) those of 3b,7b-dihydroxy-5-androsten-17-one, creatinine, Thr-Leu, triiodothyronine, anthranilic acid, pyrogallol, and cAMP. The complete list of difference-rich metabolites is shown in Table 5.

Moreover, 124 up-regulated metabolites and 107 down-regulated metabolites were identified in psoas major muscle between DLY and BC pigs. Pig breed effects (*p* < 0.05) on fatty acyls in CB rather than DLY included elevated palmitoylcarnitine, 5-OxoETE, 13-HPODE, oleic acid, 13Z,16Z-docosadienoic acid, palmitic acid, elaidic acid, stearic acid, L-palmitoylcarnitine, erucic acid, arachidic acid, and nonadecanoic acid, reduced 10-nitrolinoleate, propionylcarnitine, trans-2-butene-1,4-dicarboxylic acid, azelaic acid, and 2-methylpentanedioic acid. Additionally, the psoas major muscle from CB pigs increased (*p* < 0.05) contents of gamma-glutamyltyrosine, gamma-glutamylglutamine, L-asparagine, L-serine, Glu-Gln, and adenylosuccinic acid, while decreasing (*p* < 0.05) those of Ala-Ile, Gly-Phe, gamma-Glu-Leu, Phe-Phe, phenylacetylglycine, cysteinylglycine, Val-Ser, gamma-glutamylglutamic acid, methylmalonic acid, styrene, anthranilic acid, ADP-ribose, inosine 5′-monophosphate, cAMP, adenosine diphosphate ribose, and 3′-dephospho-CoA. The complete list of difference-rich metabolites is shown in Table 6.

### 3.5. Pathway Analysis of Differential Metabolites

Functional analysis of pathways associated with the differentially rich metabolites was performed using the KEGG database. Based on the different metabolites in longissimus dorsi muscle between the DLY-LD and CB-LD, which were associated with 72 metabolic pathways. A further pathway topology analysis was conducted and found that pig breed had a significant effect on galactose metabolism, inositol phosphate metabolism, and aldosterone-regulated sodium reabsorption (Figure 5A). As shown in Figure 5B, the different metabolites in the psoas major muscle between the DLY-PM and CB-PM were associated with 58 metabolic pathways. Among them, pig breeds had a greater impact on substance metabolism. For example, purine metabolism, caffeine metabolism, choline metabolism in cancer, and fatty acid metabolism were the pathways that were significantly affected.

### 3.6. Correlations between Meat Quality Characteristics and the Muscles Metabolites

Correlations between meat quality characteristics and the metabolites identified in *longissimus dorsi* samples are shown in Figure 6A. In general, pH _45 min_/pH _24 h_ showed high negative correlations with inositol, D-Erythrose 4-phosphate, and D-Fructose 6-phosphate in the *longissimus dorsi* muscle samples; however, for arachidic acid, erucic acid, ergocalciferol, and 5′-Adenylic acid, the correlations to pH _45 min_/pH _24 h_ were high positive. For marbling score and meat color score, poor positive correlations were found with 13-HPODE, fenpropimorph, hexadecanamide, and 5′-Adenylic acid in the meat extraction samples, whereas very high negative correlations were found with ergocalciferol and cortisone.

In Figure 6B, an overview of the relationship between the meat quality characteristics and the metabolites identified in the *psoas major* samples is displayed. Generally positive correlations were found between pH _45 min_/pH _24 h_ and the L-serine, 13Z,16Z-Docosadienoic acid, nonadecanoic acid, L-Asparagine, oleic acid, 13-HPODE, 5-OxoETE, palmitic acid, and stearic acid identified in the meat samples, whereas negative correlations were found between pH _45 min_/pH _24 h_ and the L-Fucose, 3′-Dephospho-CoA, Ala-Ile, Phe-Phe, Val-Ser, and trans-2-Butene-1,4-dicarboxylic acid identified in the meat samples. Additionally, the L*_24 h_/L*_45 min_ showed positive correlations to D-Gluconic acid, Gly-Phe, gamma-glutamylglutamic acid, L-(+)-Arabinose, inositol, methylmalonic acid, styrene, and 2-Methylpentanedioic acid and negative correlations to L-Asparagine, corticosterone, and tetrahydrocorticosterone in the meat extractions.

## 4. Discussion

With the rapid growth of consumer demand for better-quality pork in recent years, improving meat quality by breeding has been widely used in the pig industry [3,6]. Indigenous pig breeds in China are gradually favored by consumers for their advantages in meat quality [15]. In addition, exploring the characteristics of muscle fibers has gradually become one of the most effective means to investigate meat quality. Chuanzang black pig is a new variety bred in 2014 in China that has strong disease resistance and high production performance, but so far there has been no research on its meat quality. Therefore, this study selected DLY and CB pigs as research objects to compare their growth performance, carcass characteristics, meat quality traits, muscle fiber characteristics, and meat metabolite profiles.

Growth performance and carcass characteristics were known as important economic indexes of pig production [2]. Here, we noticed that CB pigs had a lower final weight and ADG and a higher F/G than DLY pigs, indicating that CB pigs had an inferior growth rate and feed conversion efficiency. Furthermore, we found that CB pigs had decreased carcass length and EMA and a tendency to reduce carcass weight compared to DLY pigs, which suggests a disadvantage of the CB pig in carcass performance that might be related to the lower growth rate of CB pigs. The metabolic status of finishing pigs could be reflected by the blood biochemical parameters. As a marker of dyslipidemia, cholesterol is moved through the bloodstream in the form of lipoprotein particles, assisted by triglycerides. The transport of cholesterol from the serum to the cell is regulated by low-density lipoproteins (LDL-C), whereas high-density lipoproteins (HDL) play an important role in the efficient reverse delivery system of cholesterol [31]. In this study, we found that the concentrations of serum CHO, LDL, and TG were lower in CB pigs than in DLY pigs, which suggests that CB pigs have a higher fat deposition capacity than DLY pigs.

Visceral indexes reflect the body’s health status, and we found that CB pigs had lower heart index, liver index, and pancreas index than DLY pig in this study, which might be explained by the poorer carcass traits of CB pigs. However, an increased kidney index was observed in the CB group. The increased kidney index may be related to the differences in renal energy metabolism, excretion function, and protein deposition between the two breeds, and the specific mechanisms need to be further explored.

Sensory quality measurement is one of the major methods to judge meat quality [2]. In this study, we found that CB pigs had better meat quality than DLY pigs. To be specific, CB pigs had a higher pH_45 min_, pH_24 h_, a*_45 min_, marbling score, and meat color score and a decreased trend of water loss rate than DLY pigs in LD muscle. Interestingly, the values of a* decrease with time for CB while increasing for DLY in PM, while the values of a* increase with time for CB and DLY, with a pronounced increase for DLY in LD. The reasons behind this phenomenon are still unclear and need to be further explored. Furthermore, CB pig also had an increased pH_45 min_, pH_24 h_, and meat color score and a lower L*_45 min_, L*_24 h_, and b*_24 h_ and cooking loss in PM muscle than DLY pig. These results suggest the advantages of CB pigs in acidity value, water-holding capacity, meat color, and intramuscular fat content. The ultimate muscle pH is one of the most important factors affecting meat quality, and the reason why the ultimate muscle pH can indirectly affect the water holding capacity and meat color of pork is that the rapid pH fall in early postmortem will cause more drips to be discharged from muscle fiber bundles [32,33]. In addition, the pH value is positively linked with glycogen content and the pH reduction rate in muscle [2,34,35]. Thus, we speculated that CB pigs might have a lower glycogen content and pH reduction rate, resulting in a higher pH value in LD and PM muscles. Meat color and water-holding capacity are closely related to pH values. Generally, a lower pH value is closely related to lighter-colored products and products with poorer water-holding capacity [36,37]. Consequently, the higher pH value might be the reason for better meat color and water loss rate in LD muscle and satisfactory meat color and cooking loss in PM muscle in CB pigs.

It is well known that meat quality is closely linked to the skeletal muscle fiber, which consists of four types: MyHC I, MyHC IIa, MyHC IIb, and MyHC IIx [8,35]. Among which, type I muscle fibers have a finer diameter, a smaller cross-sectional area, and a larger density, and their content is positively correlated with meat quality, while type II muscle fibers have a thicker diameter, a larger cross-sectional area, and a smaller density, and their content is negatively correlated with meat quality [38]. Thus, we detected the skeletal muscle histomorphology by the HE staining method. Here, we noticed that CB pigs had a finer diameter, a smaller area, and a larger density in LD and PM muscles; therefore, we speculate that CB pigs might have a higher percentage of type I muscle fibers.

RT-qPCR was used for further exploration of myofiber-type composition in LD and PM muscles. As expected, we found that CB pigs had higher mRNA levels of *MyHC I* in LD muscle and PM muscle in this trial, which confirms the results of skeletal muscle histomorphology and explains the better pH value, marbling score, meat color, and water-holding capacity in CB pigs. Furthermore, *MyHC IIx* mRNA levels in LD muscle and *MyHC IIb* and *MyHC IIx* mRNA levels in PM muscle were decreased in the CB group. It suggests that the skeletal muscle fibers of CB pigs may have been converted from type II to type I, resulting in better meat quality in CB pigs. Thus, we further examined the mRNA levels of factors associated with the transformation of muscle fiber types.

The transcription level of Forkhead box 1 (FoxO1) in type II muscle fibers is higher than that in type I muscle fibers, and its overexpression could decrease the percentage of type I muscle fibers [39,40,41]. In contrast, the transcription level of PPARδ in type I muscle fibers is higher than that in type II muscle fibers, and its activation contributed to increasing the number of type I fibers in skeletal muscle [42,43,44]. The MEF2C transcription factor could selectively activate the gene expression of type I myofibers and was reported to participate in the conversion from type II fibers to type I fibers [45,46]. NFATC1, a cofactor of MEF2, is involved in inducing skeletal muscle fiber type conversion from type II to type I [46,47]. AMPK, a major mediator that regulates skeletal myofiber type transformation, could bind to downstream genes via MEF2 to regulate the gene expression of type I fibers [7,48]. Transcription factor T-box 15 (TBX15) is intensively expressed in MyHC IIb and MyHC IIx, and its ablation can activate AMPK [49]. Here, we found that CB pigs had higher mRNA expression levels of *PPARδ*, *MEF2C*, and *NFATC1* in LD muscle and higher mRNA expression levels of *AMPKα1*, *MEF2C*, and *NFATC1* in PM muscle. It suggests that skeletal muscle type transformation in CB pigs may be mediated by AMPK, MEF2C, NFATC1, and PPARδ.

Recently, consumers have become increasingly concerned about the composition and nutritional value of meat. Therefore, consumers tend to choose higher-quality meat products that are rich in bioactive ingredients that promote health, such as unsaturated fatty acids, vitamin B1, and sterols [50]. Here, we identified metabolite composition in the *psoas major* and *longissimus dorsi* of CB and DLY pigs by LC-MS/MS. The identified metabolites were listed in Table 5 and Table 6. As described, there was a strong difference in the levels of fatty acyl compounds (e.g., 13Z,16Z-Docosadienoic acid, erucic acid, 13-HPODE, and elaidic acid), which were increased in the CB pork compared with the DLY pork. 13Z,16Z-Docosadienoic acid is a member of the omega-6 unsaturated fatty acid family, which plays a key role in human health. Anna Goc et al. (2019) observed that 13Z,16Z-Docosadienoic acid has potent bacteriostatic and bactericidal effects, which are beneficial for the preservation of meat products [51]. Along similar lines, monounsaturated erucic acid (omega-9, C22) also showed antibacterial activity in the above study [48]. With regard to 13-hydroperoxy-9,11-octadecadienic acid (13-HPODE), as a primary product of 9Z,12Z-linoleic acid, it was reported to increase the synthesis of various antioxidant enzymes, including superoxide dismutase and glutathione peroxidase, which act as inhibitors of protein and lipid oxidation [52]. Interestingly, although previous studies have demonstrated that elaidic acid may not be as efficient at promoting cholesterol excretion from cells through increased LDL receptor activity [53], generally positive correlations were found between meat quality and elaidic acid, 13-HPODE, erucic acid, and 13Z,16Z-Docosadienoic acid. Unsaturated fatty acids (UFA) are generally considered to have a higher stability in the oxidation of protein and lipids than saturated fatty acids, thus ensuring better meat quality [54]. In longissimus dorsi and psoas major, the fatty acid-related KEGG enrichment pathways were synthesis of unsaturated fatty acids and fatty acid metabolism, respectively, which is consistent with the view that Chinese indigenous pig muscle is rich in unsaturated fatty acids [15].

In addition to fatty acid metabolism, the most significant and influential different metabolomic pathways in CB pork differentiation from DLY pork were related to energy metabolism and purine metabolism. Higher levels of central carbon metabolism-related products such as D-Fructose 6-phosphate, Inositol, Dulcitol, Trehalose 6-phosphate, and L-Fucose could explain why DLY pork has a low PH value because PH is a rich source of various carbohydrates [55]. Inositol, as an energy metabolite, is not only recognized as an essential nutrient for mammalian health but is also thought to mimic insulin signaling and stimulate protein synthesis [56,57], which explains why the growth performance of DLY pigs is higher than that of CB pigs. The flavor of pork is one of the important intrinsic characteristics used to judge the quality of pork. Inosine 5′-monophosphate (IMP), as a product of ATP degradation caused by the postmortem muscle energy metabolism, is related to an umami taste [58]. Furthermore, peptides composed of different amino acids may present unique flavors because the different taste amino acids, such as “meat-like” (methionine), sour taste (alanine, tyrosine), or umami (glutamate), are the predominant components [59]. In this study, CB pork had an increased level of IMP in LD muscle compared to DLY pork. Conversely, DLY pork had an increased level of IMP, Ala-Ile, Gly-Phe, Gamma-Glu-Leu, Phe-Phe, and Val-Ser in PM muscle than CB pork, which suggested that the flavor of pork is influenced by various intrinsic factors of muscle raw material, such as breed or muscle type.

## 5. Conclusions

In sum, we present the first evidence that Chuanzang black (CB) pigs have better meat quality but poorer growth performance and carcass traits than DLY pigs. Moreover, we found that CB pigs had a higher proportion of type I oxidized fibers in skeletal muscle compared with DLY pigs, which may be mediated by AMPK, MEF2C, NFATC1, and PPARδ. Furthermore, CB pigs had higher unsaturated fatty acids but a lower abundance of central carbon metabolism-related products and peptides in muscle than DLY pigs. Given the close relationship between the traits of growth and carcass and economic benefits, improving the growth rate and carcass performance of CB pigs will be the focus of future research.

## Figures and Tables

**Figure 1 foods-12-03476-f001:**
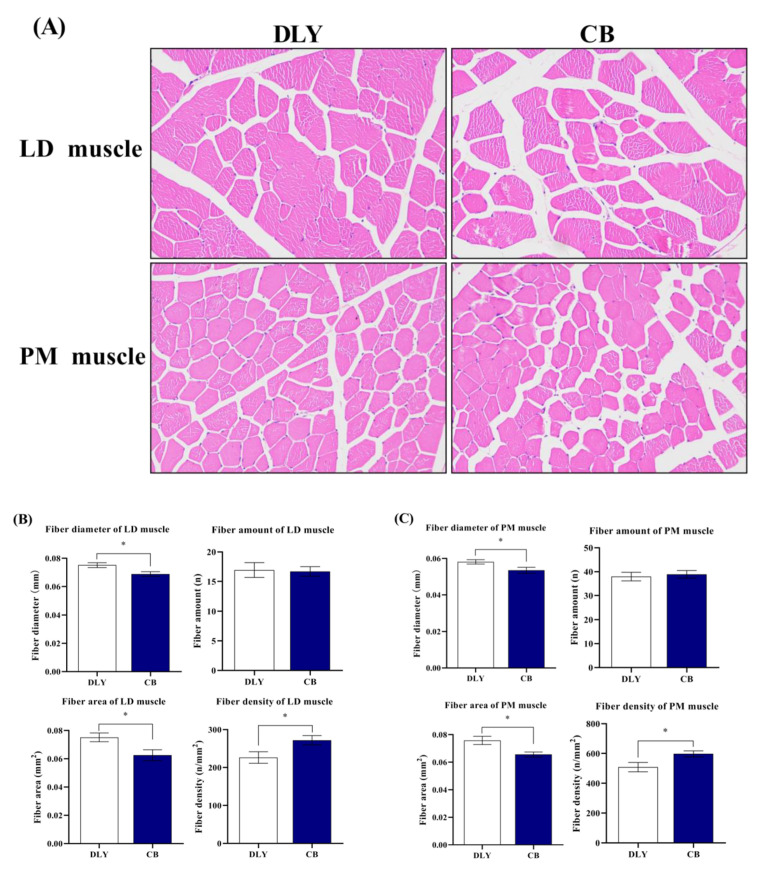
Skeletal muscle histomorphology. (**A**) Representative sections from the DLY and CB groups by hematoxylin and eosin (H&E) staining. (**B**) Comparison of fiber diameter, amount, area, and density of the LD muscle sections between the DLY and CB groups. (**C**) Comparison of fiber diameter, amount, area, and density of the PM muscle sections between the DLY and CB groups. All the histograms are presented as the mean and standard error of the means (SEM) (*n* = 10). * indicates *p* < 0.05. Abbreviations: DLY—Duroc × Landrace × Yorkshire pig; CB—Chuanzang black pig; LD muscle—longissimus dorsi muscle; PM muscle—psoas major muscle.

**Figure 2 foods-12-03476-f002:**
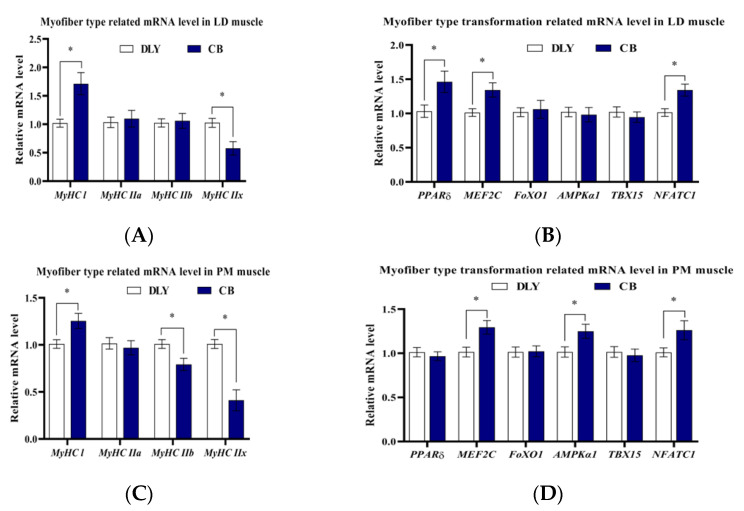
Comparison of myofiber-related mRNA levels in skeletal muscle between the DLY and CB groups. (**A**) Comparison of myofiber type-related mRNA levels in LD muscle between the DLY and CB groups. (**B**) Comparison of myofiber type transformation-related mRNA levels in LD muscle between the DLY and CB groups. (**C**) Comparison of myofiber type-related mRNA levels in PM muscle between the DLY and CB groups. (**D**) Comparison of myofiber type transformation-related mRNA levels in PM muscle between the DLY and CB groups. Data are presented as the mean and standard error of the means (SEM) (*n* = 10). * indicates *p* < 0.05. Abbreviations: DLY—Duroc × Landrace × Yorkshire pig; CB—Chuanzang black pig; LD muscle—longissimus dorsi muscle; PM muscle—psoas major muscle.

**Figure 3 foods-12-03476-f003:**
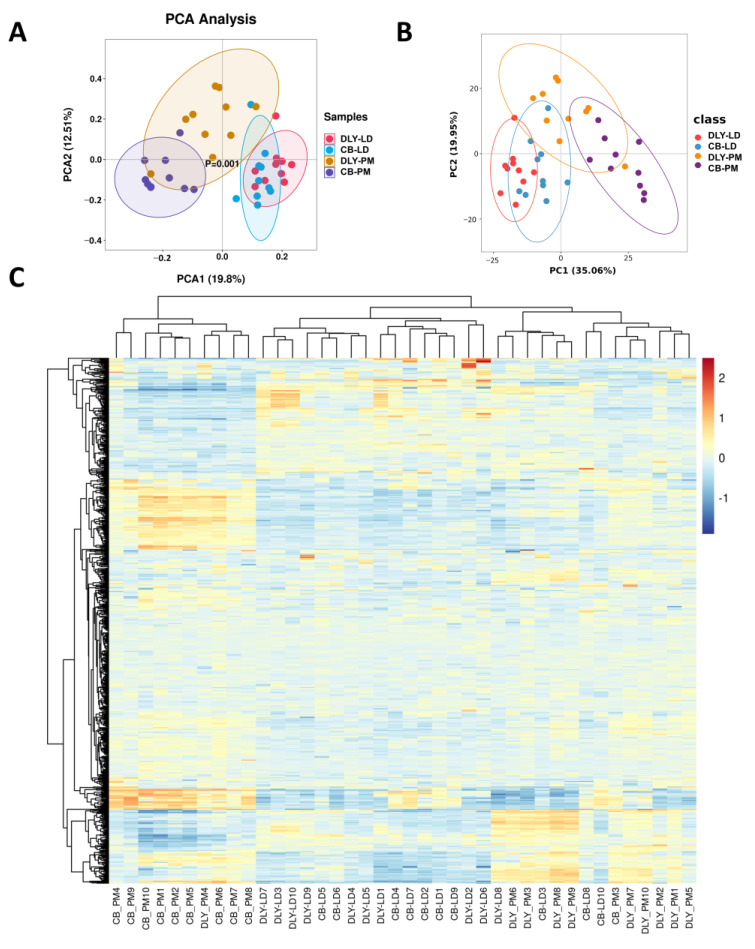
Muscle metabolome analysis. (**A**) Principal components analysis (PCA) of the longissimus dorsi and psoas major muscle metabolites from Duroc × Landrace × Yorkshire (DLY), and Chuanzang black (CB) pigs (*n* = 10). (**B**) Partial Least Squares Discrimination Analysis (PLS-DA) of the longissimus dorsi and psoas major muscle metabolites from DLY and CB pigs. (**C**) Hierarchical cluster analysis of metabolites in the longissimus dorsi and psoas major muscles of DLY and CB pigs. Heat map representation of metabolites that differed significantly between longissimus dorsi and psoas major muscle samples. Each block refers to the abundance of one metabolite from one sample.

**Figure 4 foods-12-03476-f004:**
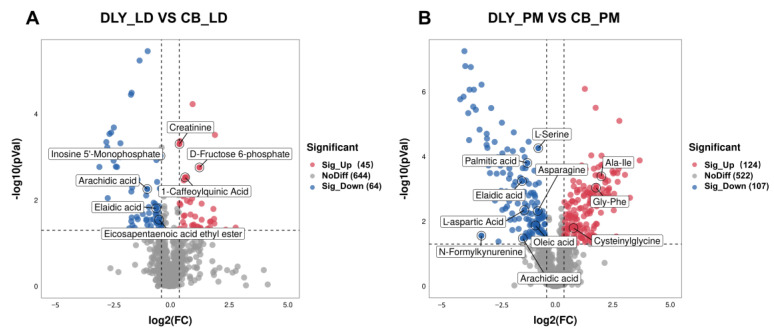
(**A**) Volcano plot presenting the significant variables in the discrimination of longissimus dorsi muscle metabolites from Duroc × Landrace × Yorkshire (DLY), and Chuanzang black (CB) pigs; (**B**) Volcano plot presenting the significant variables in the discrimination of psoas major muscle metabolites from DLY and CB pigs. The metabolites that are not significantly differential are represented as gray squares; the variables that increase significantly are indicated by a red sphere; the variables that are significantly reduced are represented by a blue sphere. The red and blue spheres in the volcano plots are metabolites for model separation following the conditions of a *p*-value of *t*-test < 0.05, variable importance in projection (VIP) > 1, and fold change (FC) ≥ 1.2, or FC ≤ 0.833.

**Figure 5 foods-12-03476-f005:**
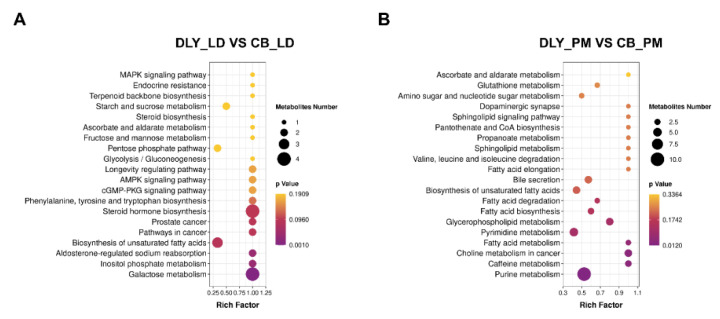
The differential metabolites were further elucidated by the KEGG signaling pathways, and the top 20 KEGG pathways of DLY-LD vs. CB-LD and DLY-PM vs. CB-PM comparisons were displayed. (**A**) Topological analysis of metabolic signaling pathways identified in longissimus dorsi muscle (*n* = 10) metabolites from Duroc × Landrace × Yorkshire (DLY), and Chuanzang black (CB) pigs; (**B**) Topological analysis of metabolic signaling pathways identified in psoas major muscle (*n* = 10) metabolites from DLY and CB pigs. The advanced bubble chart shows the degree of differentially rich metabolites in signaling pathways. The x-axis presents the rich factor (rich factor = the enrichment of different metabolites concentrated in the KEGG pathway/the enrichment of all metabolites in the background metabolites data set). The y-axis presents the concentrated KEGG pathway. The size of the bubble indicates the amount of differential abundance metabolites enriched in this pathway, and the color indicates the significance of enrichment.

**Figure 6 foods-12-03476-f006:**
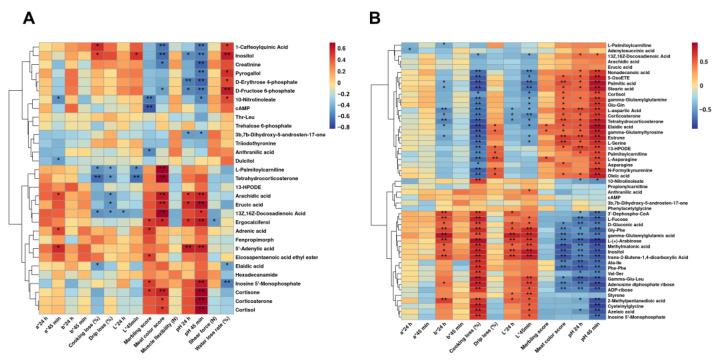
(**A**) Pearson’s correlations between meat quality characteristics and metabolites identified in the longissimus dorsi muscle (*n* = 10) from Duroc × Landrace × Yorkshire (DLY) and Chuanzang black (CB) pigs; (**B**) Pearson’s correlations between meat quality characteristics and metabolites identified in the psoas major muscle (*n* = 10) from DLY and CB pigs. * indicates *p* < 0.05; ** indicates *p* < 0.01.

**Table 1 foods-12-03476-t001:** Comparison of the growth performance between DLY and CB pigs.

Items	DLY	CB	SEM	*p*-Value
Initial weight (kg)	60.18	60.28	0.18	0.788
Final weight (kg)	128.89	120.26	1.53	<0.01
ADG (kg/pig/day)	0.92	0.80	0.02	<0.01
ADFI (kg/pig/day)	2.93	2.97	0.04	0.648
F/G	3.20	3.71	0.09	<0.01

Note: All traits in this table were analyzed with a pen as the experimental unit and presented as the mean and standard error of the means (SEM) (*n* = 50). Abbreviations: ADG—average daily gain; ADFI—average daily feed intake; F/G—feed-to-gain ratio.

**Table 2 foods-12-03476-t002:** Comparison of the serum biochemical indexes between DLY and CB pigs.

Items	DLY	CB	*p*-Value
TG	0.42 ± 0.12	0.32 ± 0.10	0.091
CHO	2.84 ± 0.22 ^a^	2.23 ± 0.45 ^b^	0.004
HDL	1.34 ± 0.16 ^a^	0.94 ± 0.28 ^b^	0.003
LDL	1.35 ± 0.17 ^a^	1.12 ± 0.25 ^b^	0.047
GLU	6.35 ± 0.75	5.44 ± 1.02	0.061
GSP	1.76 ± 0.09	1.75 ± 0.09	0.728

Note: All traits in this table were analyzed with a pen as the experimental unit and presented as the mean and standard error of the means (SEM) (*n* = 10). ^a,b^ Values without the same letters within the same line indicate a significant difference (*p* < 0.05). Abbreviations: TG—triglyceride; T-CHO—total cholesterol; HDL-C—high-density lipoprotein cholesterol; LDL-C—low-density lipoprotein cholesterol; GLU—glucose; GSP—glycosylated serum protein.

**Table 3 foods-12-03476-t003:** Comparison of carcass traits between DLY and CB pigs.

Items	DLY	CB	SEM	*p*-Value
Carcass traits				
Carcass weight (kg)	103.12	93.36	2.76	0.086
Dressing percentage (%)	79.14	77.48	1.96	0.686
Carcass length (cm)	84.11 ^a^	75.18 ^b^	1.18	<0.01
Backfat thickness (cm)				
First rib	5.50	5.31	0.18	0.597
Last rib	3.69	3.48	0.14	0.472
Last lumbar vertebra	3.35	3.49	0.12	0.565
Average backfat	4.18	4.09	0.12	0.732
EMA (cm^2^)	41.26 ^a^	29.55 ^b^	1.74	<0.01
Abdominal fat index (%)	2.09	2.28	0.09	0.305
Visceral indexes				
Kidney index (%)	0.34 ^b^	0.42 ^a^	0.01	<0.01
Heart index (%)	0.53 ^a^	0.40 ^b^	0.02	<0.01
Liver index (%)	2.40 ^a^	2.14 ^b^	0.06	0.024
Pancreas index (%)	0.32 ^a^	0.24 ^b^	0.02	<0.01

Note: Results are presented as the mean and standard error of the means (SEM) (*n* = 10). ^a,b^ Values without the same letters within the same line indicate a significant difference (*p* < 0.05). Abbreviations: DLY—Duroc × Landrace × Yorkshire pig; CB—Chuanzang black pig; EMA—eye muscle area.

**Table 4 foods-12-03476-t004:** Comparison of the meat quality between DLY and CB pigs.

Items	DLY	CB	SEM	*p*-Value
LD muscle				
pH_45 min_	6.44 ^b^	6.73 ^a^	0.05	<0.01
pH_24 h_	6.14 ^b^	6.40 ^a^	0.04	<0.01
L*_45 min_	46.17	45.58	0.62	0.649
a*_45 min_	5.87 ^b^	7.15 ^a^	0.31	0.033
b*_45 min_	0.56	0.85	0.21	0.507
L*_24 h_	48.97	48.77	0.66	0.880
a*_24 h_	7.41	7.93	0.29	0.383
b*_24 h_	2.07	1.90	0.17	0.630
Marbling score	1.20 ^b^	1.95 ^a^	0.14	<0.01
Meat color score	3.20 ^b^	3.65 ^a^	0.09	<0.01
Drip loss (%)	2.48	2.28	0.28	0.730
Cooking loss (%)	39.00	38.70	0.46	0.757
Water loss rate (%)	26.42	23.41	0.80	0.059
Shear force (N)	5950.73	5578.24	383.23	0.641
Muscle flexibility (N)	22525.07	21840.74	1677.40	0.845
PM muscle				
pH _45 min_	6.52 ^b^	6.73 ^a^	0.03	<0.01
pH _24 h_	6.30 ^b^	6.55 ^a^	0.05	<0.01
L*_45 min_	44.59 ^a^	38.57 ^b^	0.97	<0.01
a*_45 min_	14.73	16.15	0.46	0.192
b*_45 min_	2.98	3.24	0.25	0.670
L*_24 h_	44.95 ^a^	40.07 ^b^	0.96	<0.01
a*_24 h_	15.88	14.91	0.39	0.162
b*_24 h_	4.50 ^a^	2.59 ^b^	0.37	0.012
Marbling score	1.15	1.20	0.07	0.714
Meat color score	4.35 ^b^	4.95 ^a^	0.08	<0.01
Drip loss (%)	1.81	1.68	0.22	0.762
Cooking loss (%)	37.95 ^a^	29.59 ^b^	1.35	<0.01

Note: Results are presented as the mean and standard error of the means (SEM) (*n* = 10). ^a,b^ Values without the same letters within the same line indicate a significant difference (*p* < 0.05). Abbreviations: DLY—Duroc × Landrace × Yorkshire pig; CB—Chuanzang black pig; LD muscle—longissimus dorsi muscle; PM muscle—psoas major muscle.

**Table 5 foods-12-03476-t005:** Tentatively identified metabolites that have significant discrimination potential between DLY-LD and CB-LD.

Classification	Alignment ID	Metabolite Name	Formula	Average Rt (min)	Average *m*/*z*	VIP	Higher Metabolite Intensity
Fatty Acyls	Com_1503_pos	10-Nitrolinoleate	C_18_ H_31_ N O_4_	9.554	332.24301	1.391875933	DLY-LD
	Com_1848_pos	13-HPODE	C_18_ H_32_ O_4_	14.099	313.23676	1.52045014	CB-LD
	Com_1505_pos	Eicosapentaenoic acid ethyl ester	C_22_ H_34_ O_2_	14.225	348.28708	1.66408874	CB-LD
	Com_9_pos	Hexadecanamide	C_16_ H_33_ N O	15.029	256.26309	1.254291962	CB-LD
	Com_808_neg	13Z,16Z-Docosadienoic Acid	C_22_ H_40_ O_2_	15.713	335.29578	2.236571605	CB-LD
	Com_2618_neg	Erucic acid	C_22_ H_42_ O_2_	16.197	337.3114	2.177129815	CB-LD
	Com_1841_neg	Arachidic acid	C_20_ H_40_ O_2_	16.055	311.29575	1.66112227	CB-LD
	Com_47_neg	Elaidic acid	C_18_ H_34_ O_2_	15.015	281.2486	1.36796235	CB-LD
	Com_2710_neg	L-Palmitoylcarnitine	C_23_ H_45_ N O_4_	14.865	398.32758	2.06961347	CB-LD
	Com_241_neg	Adrenic acid	C_22_ H_36_ O_2_	14.911	331.26434	1.083285533	CB-LD
Organooxygen compounds	Com_7401_pos	5′-Adenylic acid	C_10_ H_14_ N_5_ O_7_ P	0.521	348.07025	2.817207398	CB-LD
	Com_1292_pos	D-Fructose 6-phosphate	C_6_ H_13_ O_9_ P	1.295	261.03683	2.183788827	DLY-LD
	Com_417_pos	Inositol	C_6_ H_12_ O_6_	1.279	181.07106	1.443484593	DLY-LD
	Com_2145_pos	Dulcitol	C_6_ H_14_ O_6_	1.301	183.08638	1.381627719	DLY-LD
	Com_118_neg	1-Caffeoylquinic Acid	C_16_ H_18_ O_9_	1.539	353.08679	1.719327459	DLY-LD
	Com_179_neg	D-Erythrose 4-phosphate	C_4_ H_9_ O_7_ P	1.252	199.0011	1.505253974	DLY-LD
	Com_706_neg	Trehalose 6-phosphate	C_12_ H_23_ O_14_ P	1.205	421.06442	2.139788243	DLY-LD
Steroids and steroid derivatives	Com_7675_pos	Cortisone	C_21_ H_28_ O_5_	11.027	361.20041	2.967351411	CB-LD
	Com_4382_pos	Corticosterone	C_21_ H_30_ O_4_	11.877	347.22113	2.694733591	CB-LD
	Com_2128_pos	Cortisol	C_21_ H_30_ O_5_	11.292	363.21603	2.910071005	CB-LD
	Com_11389_pos	Ergocalciferol	C_28_ H_44_ O	13.204	397.34213	1.429230194	CB-LD
	Com_664_pos	Tetrahydrocorticosterone	C_21_ H_34_ O_4_	11.905	368.27908	1.000092389	CB-LD
	Com_3720_pos	3b,7b-Dihydroxy-5-androsten-17-one	C_19_ H_28_ O_3_	13.067	305.21088	1.280434314	DLY-LD
Carboxylic acids and derivatives	Com_175_pos	Creatinine	C_4_ H_7_ N_3_ O	1.248	114.06584	2.491237318	DLY-LD
	Com_1122_pos	Thr-Leu	C_10_ H_20_ N_2_ O_4_	1.399	233.14949	1.570491809	DLY-LD
	Com_72553_pos	Triiodothyronine	C_15_ H_12_ I_3_ N O_4_	11.085	651.79559	1.515406142	DLY-LD
Benzene and substituted derivatives	Com_1383_pos	Anthranilic acid	C_7_ H_7_ N O_2_	1.352	138.05479	1.866095291	DLY-LD
	Com_1120_pos	Fenpropimorph	C_20_ H_33_ N O	16.599	304.26038	2.020321593	CB-LD
	Com_225_pos	Pyrogallol	C_6_ H_6_ O_3_	1.563	127.03893	1.105059701	DLY-LD
Purine nucleotides	Com_9657_pos	Inosine 5′-Monophosphate	C_10_ H_13_ N_4_ O_8_ P	16.993	349.05402	1.338298159	CB-LD
	Com_2179_pos	cAMP	C_10_ H_12_ N_5_ O_6_ P	1.493	330.05844	1.282233749	DLY-LD

Note: VIP—variable importance in projection.

**Table 6 foods-12-03476-t006:** Tentatively identified metabolites that have significant discrimination potential between DLY-PM and CB-PM.

Classification	Alignment ID	Metabolite Name	Formula	Average Rt (min)	Average *m*/*z*	VIP	Higher Metabolite Intensity
Fatty Acyls	Com_81_pos	Palmitoylcarnitine	C_23_ H_45_ N O_4_	13.031	400.34155	1.439142002	CB-PM
	Com_597_pos	5-OxoETE	C_20_ H_30_ O_3_	13.962	319.22421	1.081918664	CB-PM
	Com_1503_pos	10-Nitrolinoleate	C_18_ H_31_ N O_4_	9.554	332.24301	1.464664423	DLY-PM
	Com_1848_pos	13-HPODE	C_18_ H_32_ O_4_	14.099	313.23676	1.037461044	CB-PM
	Com_2177_pos	Oleic acid	C_18_ H_34_ O_2_	15.709	283.26294	1.255329731	CB-PM
	Com_227_pos	Propionylcarnitine	C_10_ H_19_ N O_4_	2.398	218.13853	1.630952432	DLY-PM
	Com_808_neg	13Z,16Z-Docosadienoic Acid	C_22_ H_40_ O_2_	15.713	335.29578	2.02287405	CB-PM
	Com_1163_neg	trans-2-Butene-1,4-dicarboxylic Acid	C_6_ H_8_ O_4_	1.389	143.0347	1.911630199	DLY-PM
	Com_45_neg	Palmitic acid	C_16_ H_32_ O_2_	14.921	255.23296	1.414632735	CB-PM
	Com_47_neg	Elaidic acid	C_18_ H_34_ O_2_	15.015	281.2486	1.278893606	CB-PM
	Com_27_neg	Stearic acid	C_18_ H_36_ O_2_	15.212	283.26422	1.322042603	CB-PM
	Com_2710_neg	L-Palmitoylcarnitine	C_23_ H_45_ N O_4_	14.865	398.32758	1.447689259	CB-PM
	Com_1308_neg	Azelaic acid	C_9_ H_16_ O_4_	3.954	187.0976	1.352170063	DLY-PM
	Com_2618_neg	Erucic acid	C_22_ H_42_ O_2_	16.197	337.3114	2.035958072	CB-PM
	Com_314_neg	2-Methylpentanedioic acid	C_6_ H_10_ O_4_	1.658	337.11325	1.427108573	DLY-PM
	Com_1460_neg	Nonadecanoic acid	C_19_ H_38_ O_2_	15.445	297.28003	1.200789926	CB-PM
	Com_1841_neg	Arachidic acid	C_20_ H_40_ O_2_	16.055	311.29575	1.461889362	CB-PM
Organooxygen compounds	Com_417_pos	Inositol	C_6_ H_12_ O_6_	1.279	181.07106	1.89031476	DLY-PM
	Com_4441_pos	N-Formylkynurenine	C_11_ H_12_ N_2_ O_4_	8.363	237.08664	1.474626165	CB-PM
	Com_141_neg	L-(+)-Arabinose	C_5_ H_10_ O_5_	1.353	149.04552	1.540664126	DLY-PM
	Com_256_neg	D-Gluconic acid	C_6_ H_12_ O_7_	1.302	195.05086	1.274939821	DLY-PM
	Com_492_neg	L-Fucose	C_6_ H_12_ O_5_	1.463	163.06108	1.329165871	DLY-PM
Steroids and steroid derivatives	Com_664_pos	Tetrahydrocorticosterone	C_21_ H_34_ O_4_	11.905	368.27908	2.052017617	CB-PM
	Com_2128_pos	Cortisol	C_21_ H_30_ O_5_	11.292	363.21603	1.433156433	CB-PM
	Com_4382_pos	Corticosterone	C_21_ H_30_ O_4_	11.877	347.22113	1.454058579	CB-PM
	Com_849_pos	Estrone	C_18_ H_22_ O_2_	14.136	288.19913	1.300778666	CB-PM
	Com_3720_pos	3b,7b-Dihydroxy-5-androsten-17-one	C_19_ H_28_ O_3_	13.067	305.21088	1.17201272	DLY-PM
Carboxylic acids and derivatives	Com_1612_pos	Ala-Ile	C_9_ H_18_ N_2_ O_3_	5.599	203.1391	1.627438461	DLY-PM
	Com_2153_pos	Gly-Phe	C_11_ H_14_ N_2_ O_3_	5.819	223.10756	1.690098742	DLY-PM
	Com_4666_pos	Gamma-Glu-Leu	C_11_ H_20_ N_2_ O_5_	5.428	261.14371	1.468624065	DLY-PM
	Com_7959_pos	Phe-Phe	C_18_ H_20_ N_2_ O_3_	8.651	313.15472	1.695959092	DLY-PM
	Com_3240_pos	gamma-Glutamyltyrosine	C_14_ H_18_ N_2_ O_6_	5.963	311.12369	1.602102015	CB-PM
	Com_447_pos	gamma-Glutamylglutamine	C_10_ H_17_ N_3_ O_6_	1.393	276.11877	1.418613319	CB-PM
	Com_1933_pos	Asparagine	C_4_ H_8_ N_2_ O_3_	1.258	116.0341	1.321099005	CB-PM
	Com_1045_pos	Phenylacetylglycine	C_10_ H_11_ N O_3_	8.425	194.08128	1.762377662	DLY-PM
	Com_934_pos	Cysteinylglycine	C_5_ H_10_ N_2_ O_3_ S	1.905	179.04845	1.609491744	DLY-PM
	Com_2007_pos	L-Asparagine	C_4_ H_8_ N_2_ O_3_	1.449	133.06062	1.414171912	CB-PM
	Com_749_neg	L-Serine	C_3_ H_7_ N O_3_	1.328	104.03523	1.554152699	CB-PM
	Com_332_neg	L-aspartic Acid	C_4_ H_7_ N O_4_	1.297	132.03011	1.316965758	CB-PM
	Com_1485_neg	Val-Ser	C_8_ H_16_ N_2_ O_4_	1.461	203.10345	1.24591355	DLY-PM
	Com_393_neg	gamma-Glutamylglutamic acid	C_10_ H_16_ N_2_ O_7_	1.203	275.08838	1.04954549	DLY-PM
	Com_62_neg	Methylmalonic acid	C_4_ H_6_ O_4_	1.255	117.01918	1.464664922	DLY-PM
	Com_336_neg	Glu-Gln	C_10_ H_17_ N_3_ O_6_	1.297	274.10464	1.059186976	CB-PM
Benzene and substituted derivatives	Com_567_pos	Styrene	C_8_ H_8_	12.195	105.06965	1.314401788	DLY-PM
	Com_1383_pos	Anthranilic acid	C_7_ H_7_ N O_2_	1.352	138.05479	1.663538257	DLY-PM
Purine nucleotides	Com_149_pos	ADP-ribose	C_15_ H_23_ N_5_ O_14_ P_2_	2.611	560.07886	1.681990182	DLY-PM
	Com_9657_pos	Inosine 5′-Monophosphate	C_10_ H_13_ N_4_ O_8_ P	16.993	349.05402	1.490088735	DLY-PM
	Com_2179_pos	cAMP	C_10_ H_12_ N_5_ O_6_ P	1.493	330.05844	1.159981164	DLY-PM
	Com_74_neg	Adenosine diphosphate ribose	C_15_ H_23_ N_5_ O_14_ P_2_	1.461	558.06396	1.545314364	DLY-PM
	Com_2240_neg	3′-Dephospho-CoA	C_21_ H_35_ N_7_ O_13_ P_2_ S	7.09	342.56744	1.258505255	DLY-PM
	Com_142_neg	Adenylosuccinic acid	C_14_ H_18_ N_5_ O_11_ P	1.231	462.06696	1.182469078	CB-PM

Note: VIP—variable importance in projection.

## Data Availability

The data used to support the findings of this study can be made available by the corresponding authors upon request.

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
