# Peer review of "Metabolomics Analysis Provides Novel Insights into the Difference in Meat Quality between Different Pig Breeds"

_foods, 2023, doi:10.3390/foods12183476_

Round 1
Reviewer 1 Report
Comments and Suggestions for Authors
The study is driven by the increasing interest and demands of the consumers towards meat with high quality and healthy profile. The aim of the study is to characterise and compare two pig breeds ( a modern cross reared worldwide and a cross obtained by traditional Chinese breeds) in regard to their growth performance and meat quality traits. To accomplish their aim, the authors have applied modern methods of analysis that guarantee reliable results.
The Introduction provides the necessary information that will convince the reader in the importance of the study. Moreover, the authors emphasize on a specific analysis and technique that they apply LC-MS/MS and emphasize on its advantages.
The key words are properly selected.
The experimental design however raise some questions. 1. The sex of the pigs is not mentioned. 2. Why the experiment starts at live weigh 60 kg and lasts 42 days? The breeds might be compared and included in the experiment from their weaning. Are the authors considering only the finishing period? This should be clearly stated.
The methods of analysis in their part of the water holding capacity ( drip loss, cooking loss and water retention) should be given with references.
The results of the study are presented with an adequate number of tables and figures.
The discussion is thorough and the references are up to date and relevant to the topic.
The conclusions are sound and derived from the results.
Comments on the Quality of English Language
Minor English language issues are detected, line 69: "determinate", the proper word is "determine", please, correct.
Author Response
Response to Reviewers
We thank the reviewer’s advice and we have done corresponding revision according to the comment. Thanks!
Reviewer 1
- The sex of the pigs is not mentioned.
Response: We thank the reviewer’s precious suggestion and we have added the gender of the pig at the appropriate places in the manuscript.
“A total of 100 healthy pigs were included in this study with 50 pigs (castrated boars) from each of the two crossbreeds, including 50 DLY pigs (average 60.18 ± 0.24 kg) and 50 CB pigs (average 60.28 ± 0.30 kg).”
- Why the experiment starts at live weigh 60 kg and lasts 42 days? The breeds might be compared and included in the experiment from their weaning. Are the authors considering only the finishing period? This should be clearly stated.
Response: We thank the reviewer’s comments.
The aim of this study was to characterize and compare the meat quality traits and meat metabolome of two pig breeds, DLY and CB. It is well known that the growth and development process of pigs includes slow growth stage, accelerated growth stage, rapid growth stage and finally gentle growth stage. Based on the cost and the growth cycle of pigs, farmers usually choose to slaughter pigs around 150 days. It takes about 100 days to raise pigs to 60 kg, and 100-150 is in the rapid growth period of pigs. Thus, we choose the experiment starts at live weigh 60 kg and lasts 42 days.
- The methods of analysis in their part of the water holding capacity (drip loss, cooking loss and water retention) should be given with references.
Response: We thank the reviewer’s comments and we have made corresponding revision according to the comments.
The value of pH, color coordinates, drip loss, cooking loss, water loss, marbling score, and shear force in meat, was evaluated as previously outlined [1, 2].
- Rossi, R.; Pastorelli, G.; Cannata, S.; Tavaniello, S.; Maiorano, G.; Corino, C. Effect of long term dietary supplementation with plant extract on carcass characteristics meat quality and oxidative stability in pork [J]. Meat Science 2013, 95, 542–548. https://doi.org/1016/j.meatsci.2013.05.037
- Stein, H.H.; Everts, A.K.; Sweeter, K.K.; Peters, D.N.; Maddock, R. J.; Wulf, D. M.; Pedersen, C. The influence of dietary field peas (Pisum sativum L.) on pig performance, carcass quality, and the palatability of pork [J]. Journal of Animal Science 2006, 84, 3110–3117. https://doi.org/10.2527/jas.2005-744

Reviewer 2 Report
Comments and Suggestions for Authors
In the following manuscript, the authors tried to find differences in meat quality between two different pig breeds, local (CB) and conventional (DLY). China is the biggest pork producer and meat importer in the world. Thus rationale of the study is clear and timely. This is a very comprehensive study, including essential criteria, indicators, and parameters of quality, conducted by state-of-the-art methodology. The quality of the presentation satisfies scientific criteria and is in accordance with the aim and objectives of the study. However, some parts of the manuscript must be improved as follows. Considering that China produces approximately 50% of the pork meat in the world, the economic benefits for producers and the meat industry of this research in section Conclusion must be explained and improved in that manner.
Skeletal muscle histomorphology, please insert an appropriate citation.
Table 2. Comparison of the serum biochemical indexes between DLY and CB pig. P-value must be explained as before. Are there any differences?
L241-244, The dressing percentage, backfat thickness, and abdominal fat index were not significantly affected by breed, but carcass length (P Ë‚ 0.05), EMA (P Ë‚ 0.05), heart index (P Ë‚ 0.05), liver index (P Ë‚ 0.05), pancreas index (P Ë‚ 0.05), and carcass weight (P = 0.086) were lower in CB than DLY pigs. Please rewrite (P = 0.086).
Author Response
Response to Reviewers
We thank the reviewer’s advice and we have done corresponding revision according to the comment. Thanks!
Reviewer 2
- Skeletal muscle histomorphology, please insert an appropriate citation.
Response: We thank the reviewer’s comments and we have made corresponding revision according to the comments.
The histomorphology of myofibers in LD and PM samples was measured as previ-ously outlined [1]
- Li Y.H.; Liu Y.Y.; Li, F.N.; Lin, Q.; Dai, Q.Z.; Sun, J.B.; Huang, X.G.; Chen, X.A.; Yin, Y.L. Effects of dietary ramie powder at various levels on carcass traits and meat quality in finishing pigs [J]. Meat Sci 2018, 143(undefined), 52-59. doi:10.1016/j.meatsci.2018.04.019
- Table 2. Comparison of the serum biochemical indexes between DLY and CB pig. P-value must be explained as before. Are there any differences?
Response: We thank the reviewer’s comments and we have made corresponding revision according to the comments.
The metabolic status of finishing pigs could be reflected by the blood biochemical parameters. As a marker of dyslipidemia, cholesterol is moved through the bloodstream in the form of lipoprotein particles, assisted by triglycerides. The transport of cholesterol from the serum to the cell is regulated by low-density lipoproteins (LDL-C), whereas high-density lipoproteins (HDL) play an important role in efficient reverse delivery system of cholesterol [1]. In this study, we found that the concentration of serum CHO, LDL and TG were lower in CB than in DLY pigs, which suggests that CB pigs have a higher fat deposition capacity than DLY pigs.
- Vance, J.E.; Vance, D.E. Biochemistry of Lipids, Lipoproteins and Membranes (Fifth Edition). Amsterdam: Elsevier Science 2008. https://doi.org/10.1016/B978-044453219-0.50001-5
- L241-244, The dressing percentage, backfat thickness, and abdominal fat index were not significantly affected by breed, but carcass length (P Ë‚ 0.05), EMA (P Ë‚ 0.05), heart index (P Ë‚ 0.05), liver index (P Ë‚ 0.05), pancreas index (P Ë‚ 0.05), and carcass weight (P = 0.086) were lower in CB than DLY pigs. Please rewrite (P = 0.086).
Response: We thank the reviewer’s precious suggestion and we have rewrite it.

Reviewer 3 Report
Comments and Suggestions for Authors
The paper is very interesting and well writing. It valorize Chuanzang black (CB) pig as an alternative to the international known and used breed Duroc × Landrace × Yorkshire (DLY) crossbred pig based on its meat quality, growth performance and carcass traits. However, there are some minor issues that should be adrressed to strengthen the quality of the paper.
Line 61, clarify what are included in "high production performance"
Line 84, do the authors used pigs of both sexes to run experimentation? clarify.
Line 160, "The muscle flexibility test was pretreated in the same way as the shear force analysis and measured using a texture analyzer (TA.XT. Plus, Stable Micro Systems)." A test cannot be pretreated, revise the sentence. Also, there is no need to add equipment reference as it was already mentioned above. Provide references to the methods used to assess meat quality.
Line 179, For LC-MS analysis, centrifugation alone is not sufficient to remove all debris, a filtration step is generally required.
Line 223, change CHO to GSP
Line 263, the figures’ title should be under the graph.
Line 271, table 4. PM: a* decreases with time for CB while it increases for DLY
LD: a* increases with time for CB and DLY with a pronounced increase for DLY.
No clarifications to this in the discussion section was provided. Please, revise.
Line 342, add table 5 footnote. VIP? the way of writing the chemical formula of identified compounds should be improved (also in table 6).
Line 369 change DLY-LD to DLY-PM and CB-LD to CB-PM.
Line 423-427, which factors influence feed conversion efficiency in animal? use it as discussion.
Line 431-432, might be the increase kidney could be the issue that affect carcass performance of CB. check.
Line 433, high pH (generally above 6) result in dark cutting meat, coarse texture, reduced shelf life… why do the authors consider the high pH value of CB pig as advantage that improve meat quality?
Line 446, provide studies in the literature that support the good quality of pig meat based on high pH values.
From references 6 to 53, check the numbering.
ref 18, 26, 48, 50, 52 and 53, revise according to journal requirement.
Author Response
Response to Reviewers
We thank the reviewer’s advice and we have done corresponding revision according to the comment. Thanks!
Reviewer 3
- Line 61, clarify what are included in "high production performance"
Response: We thank the reviewer’s comments and we have made corresponding revision according to the comments.
“which has strong disease resistance and high fertility”
- No Line 84, do the authors used pigs of both sexes to run experimentation? clarify.
Response: We thank the reviewer for the precious suggestion and we have added the sex of the pig in the manuscript.
“A total of 100 healthy pigs were included in this study with 50 pigs (castrated boars) from each of the two crossbreeds, including 50 DLY pigs (average 60.18 ± 0.24 kg) and 50 CB pigs (average 60.28 ± 0.30 kg).”
- Line 160, "The muscle flexibility test was pretreated in the same way as the shear force analysis and measured using a texture analyzer (TA.XT. Plus, Stable Micro Systems)." A test cannot be pretreated, revise the sentence. Also, there is no need to add equipment reference as it was already mentioned above. Provide references to the methods used to assess meat quality.
Response: We thank the reviewer’s comments and we have made corresponding revision according to the comments.
- Line 179, For LC-MS analysis, centrifugation alone is not sufficient to remove all debris, a filtration step is generally required.
Response: We thank the reviewer’s comments and we have made corresponding revision according to the comments.
“The samples were dissolved in 200 μL of isopropanol and filtered through a 0.22-μm membrane to obtain the samples that was then transferred to an LC-MS sampling bottle with an inner liner for LC-MS analysis.”
- Line 223, change CHO to GSP
Response: We thank the reviewer’s comments and we have made corresponding revision according to the comments.
- Line 263, the figures’ title should be under the graph.
Response: We thank the reviewer’s comments and we have made corresponding revision according to the comments.
- Line 271, table 4. PM: a* decreases with time for CB while it increases for DLY LD: a* increases with time for CB and DLY with a pronounced increase for DLY. No clarifications to this in the discussion section was provided. Please, revise.
Response: We thank the reviewer’s comments and we have made corresponding revision according to the comments.
- Line 342, add table 5 footnote. VIP? the way of writing the chemical formula of identified compounds should be improved (also in table 6).
Response: We thank the reviewer’s comments and we have made corresponding revision according to the comments.
- Line 369 change DLY-LD to DLY-PM and CB-LD to CB-PM.
Response: We thank the reviewer’s comments and we have made corresponding revision according to the comments.
- Line 423-427, which factors influence feed conversion efficiency in animal? use it as discussion
Response: We thank the reviewer’s comments and we have made corresponding revision according to the comments.
- Line 431-432, might be the increase kidney could be the issue that affect carcass performance of CB. check.
Response: We thank the reviewer’s comments.
“The increased kidney index may be related to the differences in renal energy metabolism, excretion function and protein deposition between the two breeds, and the specific mechanisms need to be further explored.”
- Line 433, high pH (generally above 6) result in dark cutting meat, coarse texture, reduced shelf life… why do the authors consider the high pH value of CB pig as advantage that improve meat quality?
Response: We thank the reviewer’s comments.
Similar to the reviewer's comments, a survey of five slaughterhouses in Spain showed that the pHu of pork was usually between 5.6 and 5.9 [1]. However, if the pHu > 6 in pork is the definition of DFD, only about 3% of carcasses are actually defined as DFD [2]. Therefore, a high pH (generally above 6) does not necessarily result in DFD meat quality characteristics such as dark cut color, rough texture, and shortened shelf life.
- Gispert, M., Faucitano, L., Oliver, M. A., Guárdia, M. D., Coll, C., Siggens, K., ... Diestre, A.(2000). A survey of pre-slaughter conditions, halothane gene frequency and carcass and meat quality in five Spanish pig commercial abbatoirs. Meat Science, 55, 97–106.
- Aaslyng Margit Dall., Hviid Marchen.(2020). Meat quality in the Danish pig population anno 2018. Meat Sci, 163(undefined), 108034. doi:10.1016/j.meatsci.2019.108034
- Line 446, provide studies in the literature that support the good quality of pig meat based on high pH values.
Response: We thank the reviewer’s comments.
The ultimate muscle pH is one of the most important factors affecting meat quality, and the reason why the ultimate muscle pH can indirectly affect the water holding capacity and meat color of pork is that the rapid pH fall in early post mortem will cause more drips to be discharged from muscle fiber bundles [1,2].
- Hamoen, J.; Vollebregt, H.; van der Sman, R. Prediction of the time evolution of pH in meat. Food Chem. 2013, 141, 2363–2372. https://doi.org/10.1016/j.foodchem.2013.04.127.
- Alnahhas, N.; Le Bihan-Duval, E.; Baéza, E.; Chabault, M.; Chartrin, P.; Bordeau, T.; Cailleau-Audouin, E.; Meteau, K.; Berri, C. Impact of divergent selection for ultimate pH of pectoralis major muscle on biochemical, histological, and sensorial attributes of broiler meat. J. Anim. Sci. 2015, 93, 4524–4531. https://doi.org/10.2527/jas.2015-9100.
- From references 6 to 53, check the numbering.
Response: We thank the reviewer’s comments and we have made corresponding revision according to the comments.
- ref 18, 26, 48, 50, 52 and 53, revise according to journal requirement.
Response: We thank the reviewer’s comments and we have made corresponding revision according to the comments.

Round 2
Reviewer 2 Report
Comments and Suggestions for Authors
Dear, considering the manuscript submitted after the first round of revision, I can conclude that the authors adopted all of the reviewer suggestions and that in the current form, the manuscript is acceptable for publication.
Sincerely